# The Effects of Active Methamphetamine Use Disorder and Regular Sports Activities on Brain Volume in Adolescents

**DOI:** 10.3390/jcm14155212

**Published:** 2025-07-23

**Authors:** Hüseyin Yiğit, Hatice Güler, Zekeriya Temircan, Abdulkerim Gökoğlu, İzzet Ökçesiz, Müge Artar, Halil Dönmez, Erdoğan Unur, Halil Yılmaz

**Affiliations:** 1Vocational Health School, Cappadocia University, Nevsehir 50420, Türkiye; anatomisth@gmail.com; 2Department of Anatomy, School of Medicine, Erciyes University, Kayseri 38039, Türkiye; hsusar@erciyes.edu.tr (H.G.); akerimg@hotmail.com (A.G.); unur@erciyes.edu.tr (E.U.); 3Department of Psychology, Faculty of Humanities and Social Sciences, Cappadocia University, Nevsehir 50420, Türkiye; drzekeriyatemircan@gmail.com (Z.T.); muge.artar@kapadokya.edu.tr (M.A.); 4Department of Radiology, School of Medicine, Erciyes University, Kayseri 38039, Türkiye; hdonmez@erciyes.edu.tr; 5Department of Anatomy, Faculty of Medicine, Ordu University, Ordu 52000, Türkiye; halilyilmaz855@gmail.com

**Keywords:** adolescent, brain volume, intracranial structures, methamphetamine, substance use

## Abstract

**Objectives:** Methamphetamine (MA) abuse during adolescence can have a significant impact on brain development. On the other hand, regular exercise is known to promote brain health and may have neuroprotective effects. The purpose of this study is to compare brain volumes in three different adolescent groups: those with active methamphetamine use disorder (MUD), adolescent athletes who regularly exercise, and healthy control adolescents. **Methods:** This MRI study involved three groups of adolescents: 10 with active MUD (9 males, 1 female), nine licensed runner adolescents (three males, six females), and 10 healthy adolescents (5 males, 5 females). Brain volumes were analyzed using T1-weighted images from a 3.0 Tesla MRI scanner, and then segmented automatically with vol2Brain. Statistical analyses included ANCOVA with sex as a covariate and LSD post hoc tests performed using SPSS Statistics 23. **Results:** Adolescents with MUD showed a 10% increase in total white matter volume compared to the athlete group. Conversely, cortical gray matter volume was reduced by 4% compared to the healthy control group and by 7% compared to the athlete group. The frontal and insular cortices in the MUD group had significantly diminished volumes compared to the athlete group. Overall, individuals with MUD had decreased gray matter volumes and increased white matter volumes in their brains. The brain volumetric differences between the MUD group and the athlete group were statistically significant. **Conclusions:** The brains of those with MUD displayed a reduction in gray matter volume and an increase in white matter volume, indicating damage from MA on the developing adolescent brain. The volumetric disparities between the MUD and athlete groups were found to be significantly different, suggesting a possible neuroprotective factor of exercise. Further studies are required to explore the potential of exercise-based interventions in alleviating the harmful effects of MA abuse.

## 1. Introduction

Methamphetamine (MA), a derivative of amphetamine, acts as a stimulant in the central nervous system (CNS). Originally used for treating obesity and depression, it has now become a global epidemic due to its addictive properties [1]. In 2023, it was identified as the second most commonly abused illicit drug worldwide, following cannabis [2]. Extensive research has focused on the effects of MA on the brain, particularly through neuroimaging studies using MRI. These studies have shown that MA can lead to structural changes in various brain regions, such as decreased gray matter volume and disruptions in white matter, the corpus striatum, and the parietal cortex. These changes have been associated with inflammation and cognitive impairments, particularly affecting learning and memory [1]. Individuals with methamphetamine use disorder (MUD) and psychosis have been found to have reduced volumes in the hippocampus and amygdala [3]. Furthermore, prenatal exposure to MA has been linked to specific brain abnormalities, including increased cortical thickness in some regions and decreased volumes in others [4].

Exercise has shown promise as an intervention for MUD, potentially facilitating MA cessation [5,6,7,8]. Animal models and clinical studies have extensively documented the neuroprotective benefits of physical activity, including enhanced memory and learning, promotion of neurogenesis, and protection against neurodegeneration [9]. Exercise promotes neuronal survival and neuroplasticity by increasing the production of neurotrophic factors, neurotransmitters, and hormones. It also exerts protective effects on the nervous system by stimulating processes such as synaptic plasticity, neurogenesis, and angiogenesis, while reducing neuroinflammation. These neuroprotective effects are mediated through enhanced cerebral blood flow, diminished inflammation, and overall improved neuroplasticity. Exercise also maintains cellular homeostasis by enhancing waste clearance systems like the glymphatic system and autophagy, augmenting antioxidant capacity, and strengthening DNA repair mechanisms [10,11]. In humans, exercise has been shown to increase brain volume in areas associated with executive function and enhance cognitive performance across various populations [12,13].

Research on the effects of MA use on the adolescent brain is limited, with most studies focusing on adults or individuals in recovery [14]. This leaves a gap in understanding the impact of active MA use on adolescent brain structure. In contrast, the positive effects of exercise on physical and mental health are well-known, but research on brain volume changes in adolescents is lacking. We hypothesize that regular exercise will have positive effects on adolescent brain structures, while active MA use will have negative effects. To test this hypothesis, an MRI study was conducted comparing brain volumes among three groups: adolescents with active MUD, adolescents engaged in regular exercise, and a control group.

## 2. Materials and Methods

### 2.1. Participant Selection

This study received approval from the Cappadocia University Local Ethics Committee under decision number 23.18, dated 27 December 2023. Prior to the study commencement, written and oral informed consent was obtained from all participants and their parents or legal guardians, in accordance with the principles of the Declaration of Helsinki and other relevant ethical guidelines.

Ten adolescents with active MA dependence (mean age: 16.6 years; 9 males, 1 female) were recruited from local drug rehabilitation centers. A comparison group of nine licensed runners (mean age: 13.5 years; 3 males, 6 females) was recruited from local sports centers. Demographic characteristics, MA use frequency and quantity, and durations of sports activities for individuals with MUD and athletes are detailed in Table 1. Additionally, ten healthy adolescents with no history of substance use disorder were also included in this study (mean age: 14.8 years; 5 males, 5 females).

Adolescents with MUD were included if they were between 12 and 17 years old, met DSM-5 criteria for MUD, had a history of regular MUD for at least 12 months, and did not have a history of other illicit drug use, psychosis, significant head injury, post-concussion symptoms, or chronic medical, neurological, or psychiatric illnesses.

Licensed athletes were eligible if they were between 12 and 17 years old, had held a valid runner’s license for at least 12 months, did not use cigarettes or alcohol, and did not have a history of psychosis, significant head injury, post-concussion symptoms, or chronic medical, neurological, or psychiatric illnesses.

All healthy control subjects had no history of substance abuse. Only one control subject had a history of smoking, which lasted less than one month. None of the individuals in the control, MUD, or exercise groups were taking psychotropic medication.

### 2.2. MRI Protocol

Prior to the MRI scan, all metallic items/devices were removed from all participants to prevent any potential harm. All participants underwent cranial MRI examination using a 3.0 Tesla superconductive MRI scanner (Ingenia, Philips). Subjects were positioned supine with a 16-channel head coil; no sedation was administered. Non-contrast, three-dimensional volumetric T1-weighted 3D volumetric images were acquired with the following parameters: field of view = 240 mm, matrix = 480, number of excitations = 1.0, slice thickness = 1 mm, spacing between slices = 0.5 mm, repetition time = 6.7, and echo time = 3.0.

### 2.3. MR Data Analysis

T1-weighted 3D volume files were converted to NIfTI format using dcm2nii (https://www.nitrc.org/projects/dcm2nii/, accessed on 5–10 January 2024) and Radiant software (https://www.radiantviewer.com/, accessed on 10 January–20 February 2024). The resulting files were uploaded to the vol2Brain website (https://www.volbrain.net, accessed on 10 January–20 February 2024) for automated segmentation of various brain structures.

### 2.4. Statistical Analysis

Statistical analyses were conducted using IBM SPSS Statistics 23. Normal distribution assumption was evaluated based on skewness, kurtosis, Standard Deviation (Std)/Mean ratio, Shapiro–Wilk test, histograms, and Q-Q plots. Data were considered normally distributed if they met at least three of these criteria [15]. Data are presented as Mean ± Std. Analysis of Covariance (ANCOVA) was performed with sex as a covariate. Partial eta-squared ηp2 values were used to determine the proportion of variance explained. Post hoc power analysis was conducted, and observed power is reported. LSD post hoc tests were used to identify significant group differences.

## 3. Results

In this study, 135 different brain regions were analyzed for cortical thickness (mm) and relative volume (mm^3^/ICV). Table 2 shows the statistical results for absolute volume values in adolescent groups with MUD, licensed athletes, and the controls.

Table 3 provides analyses of relative volumes among different groups. Adolescents with MUD had a 10% greater total white matter volume compared to athletes (*p* = 0.001). They also had 8% larger volumes in total cerebral white matter, right cerebral white matter, and left cerebral white matter compared to athletes (*p* = 0.001, *p* = 0.002, *p* = 0.001, respectively). However, adolescents with MUD showed marked reductions in brain gray matter volumes. Total gray matter volume in the MUD group was 5% lower than in athletes (*p* = 0.001). Cortical gray matter volume was 4% lower in the MUD group compared to the controls and 7% lower compared to athletes (*p* < 0.001). Similarly, total cerebral gray matter volume in the MUD group was 3% lower than in the controls and 6% lower than in athletes (*p* < 0.001). At the hemispheric level, right hemispheric gray matter volume in the MUD group was 6% smaller than in athletes (*p* < 0.001). Left hemispheric gray matter volume was 3% lower in the MUD group compared to the controls and 6% lower compared to athletes (*p* < 0.001).

In adolescents with MUD, the volume of the right triangular inferior frontal gyrus was 15% smaller than in the control group (*p* = 0.015). The right insular cortex volume was 9% smaller in the MUD group compared to athletes (*p* = 0.024), while the left insular cortex volume was 6% smaller relative to the controls and 8% smaller relative to athletes (*p* = 0.009). Adolescents with MUD also showed significant volumetric reductions in specific brain regions compared to athletes, including a 7.5% smaller left anterior insula volume (*p* = 0.011), an 11% smaller total central operculum volume (*p* = 0.006), and an 11% smaller right central operculum volume (*p* = 0.017). Additionally, total frontal operculum volume was found to be 13% smaller in the MUD group compared to athletes (*p* = 0.006). Individuals with MUD had a smaller gray matter volume but a larger white matter volume compared to the athlete group. Although no prominent differences were observed in other lobes, the frontal cortex and insular cortex volumes were significantly smaller in individuals with MUD compared to athletes.

## 4. Discussion

This study compared the effects of MUD on brain volumes in adolescents with the potential neuroprotective effects of regular sports activities. The findings showed that MUD negatively impacts adolescent brain development, leading to increased white matter volume and decreased gray matter volume. Significant volume reductions were observed in key brain regions like the frontal and insular cortices in the MUD group. In contrast, adolescents who regularly exercise did not show many of these adverse effects, highlighting the protective role of sports on brain health.

Jernigan et al. found that MA-abstinent individuals had increased volumes in the basal nuclei, particularly the nucleus accumbens, and the parietal lobe [16]. In a study by Chang et al. in MA abstinent individuals with a mean age of 32.1 years, an increase in putamen and globus pallidus volumes was observed compared to the control group [17]. In a review by Chang et al., one of the reasons for the volumetric increase in these structures was explained as increased water content in the white matter and increased inflammation in neurons [18]. Our study showed a higher white matter volume in MUD individuals compared to athletes, but no significant difference in white matter volume between MUD individuals and the control group in adolescents. This suggests that MA use may lead to an increased white matter volume compared to exercise. Additionally, our study revealed a lower grey matter volume in MUD individuals compared to licensed athletes, but no significant difference in total brain volume between the groups. This may indicate that the increase in white matter could be a compensatory response to offset the decrease in grey matter, but further evidence from functional (fMRI) and diffusion tensor imaging (DTI) studies is needed to confirm this.

Volume reductions in the basal nuclei, right frontal cortex, and left occipitoparietal cortex have been observed in children exposed to prenatal MA. Additionally, there were volumetric increases in limbic structures such as the anterior and posterior cingulate, temporal lobe, and inferior frontal gyrus [19]. Another study found an increase in left putamen volume and a decrease in cortical thickness in the opercular region of the left parietal lobe and precuneus in children exposed to prenatal MA [20]. Furthermore, prenatal MA exposure was linked to a decrease in caudate nucleus and thalamus volume in newborns [21]. In our study, individuals with MUD did not have prenatal MA exposure. In individuals with MUD but no prenatal MA exposure, the volume of the triangular inferior frontal gyrus was 15% lower than the control group, while the total frontal lobe volume was 9% lower than that of the athlete group. These findings suggest a significant association between MA exposure during adolescence and alterations in the developing brain.

In a study, it was found that there is a positive correlation between cortical thickness and intelligence quotient (IQ) [22]. Petzold et al. conducted a retrospective study that revealed that individuals who had previously used MA and ceased use exhibited thinning in bilateral frontal, parietal, temporal, insular, and right cingulate cortex thicknesses. The researchers proposed that cortical thinning post-MA use is associated with neuronal damage and may lead to volumetric expansion through glial proliferation [23]. Our results show that individuals who actively use MA have significantly less gray matter in the insular cortex, anterior insula (both right and left), frontal operculum, central operculum, and frontal lobe areas compared to the athlete group. Consistent with Petzold et al., our study showed that individuals with MUD had significantly thinner frontal, parietal, and temporal cortices compared to those who participated in exercise. This might imply that engaging in sports is connected to a higher cortical thickness, potentially reflecting typical development rather than factors like glial proliferation or neuronal damage. It also implies that glial proliferation could be regulated during adolescence, with no cortical thinning seen in the occipital, insular, and limbic cortices. Groups can also be differentiated macroscopically based on differences in cortical thickness (Figure 1, Appendix A).

Nakama and colleagues found that MA users experience greater loss of cortical gray matter compared to normal aging [24]. Our study showed that adolescents with MUD had lower cortical gray matter volume than adolescents who exercised. Both total gray matter and cortical gray matter volumes were significantly lower in adolescents with MUD compared to exercisers, suggesting increased vulnerability to gray matter deficits with age. Thompson et al. observed increased white matter volume in the lateral ventricle and decreased volume in the right medial cingulate cortex and hippocampus in individuals with MA use disorders [25]. However, our study did not find significant changes in ventricular volumes and cerebrospinal fluid (CSF) volume between the groups, indicating that impairments in CSF and ventricles may develop in later ages.

Chronic use of MA has been shown to impact the structure and function of the frontal lobe [26]. Studies have found that MA users have smaller gray matter volume in the triangular inferior frontal gyrus and reduced frontal activation during cognitive tasks [24,27,28]. There is also a link between the frontal cortex and changes in personality [29]. Increased aggression is a common outcome of MA use [30]. In this study, individuals using MA had a 9% smaller total frontal lobe volume compared to athletes. The volume of the triangular inferior frontal gyrus in MA users was 15% smaller than in the control group. Therefore, the aggressive behavior seen in individuals with MA use disorder may be related to structural changes in the frontal lobe.

Recent research has highlighted the potential therapeutic benefits of physical exercise for individuals who use MA [31]. Studies have shown that engaging in physical exercise can lead to improved cognitive function, lower relapse rates, and longer periods of abstinence in MA users [32,33]. Our study found that the differences in brain volume between the MUD group and those who regularly exercise were more significant than those between the MUD group and non-exercisers. This suggests that incorporating regular exercise into the treatment plan for MA use disorder could have a more positive impact on the therapeutic process, potentially leading to lasting behavioral changes in a shorter period of time.

Ruan et al. conducted a comparative study with individuals who had been using MA for an average of 6.3 years and had abstained for 6 months, compared to a control group of non-drug users. They found reduced gray matter volume (GMV) in several brain regions in the 6-month-abstinent MA users compared to the control group [34]. Another study by Zhang et al. compared MA users who had abstained for at least 14 months with healthy controls and found differences in GMV in various brain regions [35]. Hu et al. examined males with MUD and found smaller volumes in several brain regions compared to healthy controls [36]. In our study, we found reduced GMV in certain brain regions among individuals with MUD compared to athletes, which presents a challenge for direct comparisons with other studies that focus on abstinence.

The current study is limited in its ability to fully characterize participant-specific factors such as diet, lifestyle, IQ, and specific phenotypes. These unmeasured variables may have influenced the observed differences in brain structures, complicating the interpretation of the findings. Future research should include comprehensive assessments of these factors to better understand their potential effects on brain morphology. Additionally, this study had a small sample size with a limited number of participants who were both MUD-affected and adolescent athletes, which significantly limits the generalizability of the results. Larger studies are needed to improve the power and external validity of future research in this area. The gender distribution in this study was imbalanced, with males being the majority in the MUD group and females in the athlete group. To address this, relative brain volumes were analyzed, and gender was included as a covariate in the analysis. This study is a pilot study, and future studies should strive for a more balanced gender distribution and a larger sample size to investigate potential gender-specific differences in brain structure. Longitudinal studies that consider dietary habits, age-related changes, and specific characteristics are necessary to gain a better understanding of the effects of MUD on brain structure.

## 5. Conclusions

This study reveals important results from a comparative analysis of brain volume variances in adolescents with MUD and adolescent athletes who participate in regular exercise. Adolescents with MUD showed a notable rise in total white matter compared to the athlete group. On the other hand, cortical gray matter volume was significantly lower in the MUD group compared to both the healthy control group and the athlete group. Reductions in volume were especially noticeable in key brain regions like the frontal and insular cortex in the MUD group compared to the athlete group.

Our study proposes the hypothesis that the observed reduction in gray matter volume in individuals with MUD could potentially be counterbalanced by an increase in white matter volume; however, more detailed investigation with fMRI and DTI studies is necessary to validate this hypothesis. In the adolescent population, there were no significant differences in ventricle and CSF volumes between groups, suggesting that MA use may have more pronounced effects on these structures later in life. Notably, individuals in the MUD group exhibited a smaller total frontal lobe volume compared to athletes and a smaller triangular inferior frontal gyrus volume compared to the control group.

This study emphasizes the negative impact of MA use on adolescent brain structure and proposes that regular physical activity could offer protection. Integrating exercise into MUD treatment may enhance results. Further research should investigate the potential of exercise interventions in reducing the harmful effects of MUD, utilizing advanced neuroimaging and biomarker analysis techniques, a larger participant pool, and considering other individual factors in analyses for a more thorough understanding.

## Figures and Tables

**Figure 1 jcm-14-05212-f001:**
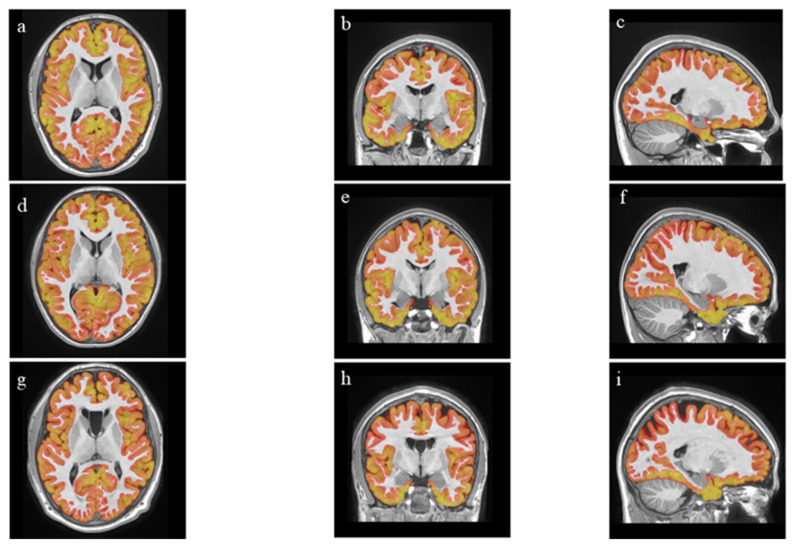
Axial (**a**,**d**,**g**), coronal (**b**,**e**,**h**), and sagittal (**c**,**f**,**i**) MR images of cortical thickness differences of adolescents with MUD, athletes, and the control group. (**a**–**c**) Control group; (**d**–**f**) athletes; and (**g**–**i**) adolescents with MUD.

**Table 1 jcm-14-05212-t001:** Detailed characteristics of licensed athletes and methamphetamine use disorder groups.

Participant ID	Sex	Age (Years)	Years of Sport Participation	Weekly Running Frequency	Weekly Running Distance	Single Training Session Duration	Years of Substance Use	Daily Substance Use Frequency	Weekly Substance Use Frequency (Days)	Average Dose per Use (mg)	Total Daily Dose (Average mg)
A1	F	13	2	3	25–35 km	60–75 min	N/A	N/A	N/A	N/A	N/A
A2	F	12	1	3	25–35 km	60–75 min	N/A	N/A	N/A	N/A	N/A
A3	M	16	3	3	25–35 km	60–75 min	N/A	N/A	N/A	N/A	N/A
A4	M	13	2	3	25–35 km	60–75 min	N/A	N/A	N/A	N/A	N/A
A5	F	16	2	3	25–35 km	60–75 min	N/A	N/A	N/A	N/A	N/A
A6	F	13	1	3	25–35 km	60–75 min	N/A	N/A	N/A	N/A	N/A
A7	F	13	1	3	25–35 km	60–75 min	N/A	N/A	N/A	N/A	N/A
A8	M	13	1	3	25–35 km	60–75 min	N/A	N/A	N/A	N/A	N/A
A9	F	13	2	3	25–35 km	60–75 min	N/A	N/A	N/A	N/A	N/A
MUD1	M	17	N/A	N/A	N/A	N/A	4	3	4	150	450
MUD2	M	17	N/A	N/A	N/A	N/A	3	2	5	150	300
MUD3	M	16	N/A	N/A	N/A	N/A	4	3	5	150	450
MUD4	M	17	N/A	N/A	N/A	N/A	5	3	6	150	450
MUD5	M	17	N/A	N/A	N/A	N/A	4	2	3	150	300
MUD6	M	17	N/A	N/A	N/A	N/A	5	3	5	150	450
MUD7	M	16	N/A	N/A	N/A	N/A	3	2	5	150	300
MUD8	M	16	N/A	N/A	N/A	N/A	3	2	4	150	300
MUD9	M	16	N/A	N/A	N/A	N/A	3	3	4	150	450
MUD10	F	17	N/A	N/A	N/A	N/A	3	2	4	150	300

A: Licensed athlete; MUD: methamphetamine use disorder; F: female; M: male; and N/A: not applicable.

**Table 2 jcm-14-05212-t002:** Comparative analysis of major intracranial structures across groups.

Intracranial Structure (mm^3^)	Control (Mean ± Std)	MUD (Mean ± Std)	Athlete (Mean ± Std)	Sig. (*p*)	(η_p_^2^)	Observed Power
White matter	494.87 ± 30.97	518.73 ± 53.41	455.26 ± 49.04	0.051	0.266	0.638
Gray matter	776.20 ± 72.27	759.72 ± 58.96	757.11 ± 89.57	0.506	0.088	0.197
Total brain volume	1271.07 ± 88.91	1278.45 ± 109.72	1212.37 ± 134.77	0.428	0.103	0.230
Intracranial cavity	1422.54 ± 96.97	1435.26 ± 132.41	1331.70 ± 145.33	0.221	0.159	0.362
Cerebrum T	1123.76 ± 78.95	1128.37 ± 105.54	1072.80 ± 121.81	0.504	0.088	0.198
Frontal lobe T	212.55 ± 18.77	199.85 ± 21.32	206.87 ± 28.38	0.219	0.159	0.364
Temporal lobe T	127.43 ± 12.09	125.10 ± 11.43	122.99 ± 15.86	0.639	0.064	0.151
Parietal lobe T	113.82 ± 12.19	112.31 ± 8.80	114.23 ± 13.96	0.974	0.009	0.061
Occipital lobe T	80.05 ± 10.06	79.44 ± 10.46	79.49 ± 11.40	0.515	0.086	0.194
Limbic cortex T	45.55 ± 4.37	45.88 ± 4.76	45.00 ± 6.27	0.720	0.051	0.127
Insular cortex T	34.24 ± 3.95	32.38 ± 3.11	32.43 ± 5.81	0.440	0.101	0.225

Data are presented as Mean ± Standard Deviation (Std). ANCOVA was performed with sex as a covariate for statistical analysis. Partial eta-squared (η_p_^2^) values are reported to indicate the proportion of variance explained. Observed power analysis is also included in the table. Post hoc comparisons were not conducted due to the absence of significant differences between groups. MUD: Participants with methamphetamine use disorder; Sig: significance; T: total.

**Table 3 jcm-14-05212-t003:** Comparison of relative brain volume in different brain structures among groups.

Brain Structure	Control (Mean ± Std)	MUD (Mean ± Std)	Athlete (Mean ± Std)	Sig. (*p*)	(η_p_^2^)	Observed Power
White matter	39.00 ± 2.07	40.52 ± 1.12	37.58 ± 1.27 ^b^	0.001	0.460	0.960
Gray matter	60.99 ± 2.07	59.47 ± 1.12	62.41 ± 1.27 ^b^	0.001	0.460	0.960
Cortical grey matter	48.21 ± 1.58	46.24 ± 1.25 ^a^	49.60 ± 1.55 ^b^	<0.001	0.527	0.991
Cerebrum WM T	36.50 ± 2.00	37.98 ± 1.14	35.14 ± 1.14 ^b^	0.001	0.458	0.959
Cerebrum WM R	18.28 ± 1.01	19.06 ± 0.60	17.65 ± 0.61 ^b^	0.002	0.447	0.950
Cerebrum WM L	18.22 ± 0.99	18.91 ± 0.54	17.48 ± 0.59 ^b^	0.001	0.454	0.956
Cerebrum GM T	51.90 ± 1.54	50.22 ± 1.11 ^a,c^	53.32 ± 1.51	<0.001	0.516	0.987
Cerebrum GM R	26.05 ± 0.78	25.22 ± 0.59 ^c^	26.77 ± 0.82	<0.001	0.476	0.971
Cerebrum GM L	25.85 ± 0.76	24.99 ± 0.53 ^a,c^	26.54 ± 0.79	<0.001	0.511	0.989
Frontal lobe T	16.70 ± 0.59	15.62 ± 0.85 ^a,c^	17.03 ± 0.98	0.001	0.460	0.961
Frontal lobe L	8.27 ± 0.32	7.73 ± 0.44 ^a,c^	8.42 ± 0.53	0.003	0.424	0.930
Triangular inf. frontal gyrus R	0.32 ± 0.03	0.27 ± 0.04 ^a,c^	0.29 ± 0.02	0.015	0.336	0.797
Insular cortex R	1.33 ± 0.10	1.25 ± 0.07 ^c^	1.38 ± 0.05	0.024	0.309	0.739
Insular cortex L	1.35 ± 0.07	1.27 ± 0.05 ^a,c^	1.38 ± 0.08	0.009	0.365	0.849
Anterior insula L	0.38 ± 0.02	0.37 ± 0.02 ^c^	0.40 ± 0.02	0.011	0.355	0.832
Central operculum T	0.71 ± 0.05	0.67 ± 0.03 ^c^	0.75 ± 0.03	0.006	0.382	0.877
Central operculum R	0.35 ± 0.02	0.33 ± 0.03 ^c^	0.37 ± 0.02	0.017	0.328	0.781
Frontal operculum T	0.38 ± 0.03	0.34 ± 0.02 ^c^	0.39 ± 0.02	0.006	0.384	0.879

Data are presented as Mean ± Standard Deviation (Std). ANCOVA was performed with sex as a covariate for statistical analysis. LSD post hoc tests were conducted to determine differences between groups. Partial eta-squared (η_p_^2^) values are reported to indicate the proportion of variance explained. Observed power analysis is also included in the table. ^a^: Statistically significant compared to the control group; ^b^: statistically significant compared to the MUD group; ^c^: statistically significant compared to the athlete group; MUD: participants with methamphetamine use disorder; Sig.: significance; WM: white matter; GM: gray matter; T: total; R: right; L: left; and inf: inferior.

## Data Availability

The data that support the findings of this study are available from the corresponding author upon reasonable request.

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
