# Peer review of "The Effects of Active Methamphetamine Use Disorder and Regular Sports Activities on Brain Volume in Adolescents"

_jcm, 2025, doi:10.3390/jcm14155212_

Round 1

Reviewer 1 Report

Comments and Suggestions for Authors

Strengths of the Paper

Novel and Significant Topic:

This study addresses a novel and clinically significant topic by examining the effects of methamphetamine (MA) use on brain structure in adolescents, and innovatively introduces physical exercise as a potential protective factor, providing a unique perspective.

Main Issues of the Paper

Significant Methodological Limitations, Particularly Regarding Control Group Design:

Substantial Group Differences in Demographic Characteristics:

  • Age: The mean ages were 16.8 years for the MUD group, 15.2 years for the exercise group, and 14.8 years for the control group. Although numerically similar, given the rapid brain development occurring during adolescence, even small differences in age (particularly the older age of the MUD group) may substantially impact the outcomes.
  • Gender Ratio: There was a major imbalance in gender distribution among groups (90% males in the MUD group, 70% females in the exercise group, and approximately balanced gender ratio in the control group). Since gender significantly affects brain structural measures (such as brain volume and gray/white matter proportions), this imbalance considerably undermines the interpretability of the results.

Questionable Appropriateness of the Exercise Group as a Control:

The exercise group consisted of enrolled athletes, whose lifestyles (regular high-intensity exercise, dietary habits, and sleep patterns) markedly differ from typical healthy adolescents. Using such a group as the primary control for the MA-dependent group severely hampers the effective separation of "exercise effects" from "MA effects," thereby reducing the internal validity of the study.

Insufficient Sample Size and Low Statistical Power:

With each group containing only around 10 participants, the sample size is critically inadequate. Such limited numbers significantly reduce the statistical power, making robust conclusions difficult to achieve even with corrections for multiple comparisons (e.g., Bonferroni correction). Thus, the reliability of the study findings remains uncertain.

Inadequate Control of Potential Confounding Factors:

The study lacks adequate control or detailed descriptions of critical variables potentially influencing brain structure, such as intelligence quotient (IQ), lifestyle factors (eating habits, sleep quality, stress levels), and detailed states prior to MRI scanning (e.g., whether the exercise group was scanned immediately post-exercise, drug metabolic conditions). These uncontrolled factors may represent significant confounders, limiting both the reliability and generalizability of the findings.

Overly Optimistic Interpretation of Results Lacking Sufficient Evidence:

The authors speculate that MA-induced reductions in gray matter and increases in white matter may reflect a "compensatory mechanism." However, this interpretation is overly speculative due to a lack of supporting neuropathological or additional neuroimaging evidence (e.g., functional imaging or diffusion imaging).

Author Response

Dear Reviewer 1,

Thank you for your valuable feedback and important suggestions, which have significantly contributed to our manuscript. We have addressed your comments point-by-point, and the corresponding changes have been highlighted in yellow within the revised manuscript.

As line numbers are not displayed in the Word version of the journal's template, we were unable to refer to specific line numbers for our explanations. Instead, all revisions and our responses are clearly detailed under each "Author's Response" heading.

We appreciate your thorough review.

Reviewer 1: Strengths of the Paper

Novel and Significant Topic:

This study addresses a novel and clinically significant topic by examining the effects of methamphetamine (MA) use on brain structure in adolescents, and innovatively introduces physical exercise as a potential protective factor, providing a unique perspective.

Main Issues of the Paper

Significant Methodological Limitations, Particularly Regarding Control Group Design:

Substantial Group Differences in Demographic Characteristics:

Age: The mean ages were 16.8 years for the MUD group, 15.2 years for the exercise group, and 14.8 years for the control group. Although numerically similar, given the rapid brain development occurring during adolescence, even small differences in age (particularly the older age of the MUD group) may substantially impact the outcomes.

Gender Ratio: There was a major imbalance in gender distribution among groups (90% males in the MUD group, 70% females in the exercise group, and approximately balanced gender ratio in the control group). Since gender significantly affects brain structural measures (such as brain volume and gray/white matter proportions), this imbalance considerably undermines the interpretability of the results.

Author’s response: Dear Reviewer 1,

Thank you for your effort in evaluating our study and contributing to its improvement. We recognize that our study has significant limitations regarding age and sex, and we've taken steps to address these concerns.

To mitigate the impact of sex, we performed ANCOVA analyses with sex as a covariate. To demonstrate the reliability of our results, we also conducted partial eta-squared and observed power analyses, which indicated a high degree of confidence in our findings.

Furthermore, to minimize the influence of both age and sex, we already included relative volumes in our analyses. We acknowledge that age is an important covariate. However, in the literature on adolescent brain studies, there are examples that consider the 10-18 age range as a single group (e.g., Batın, Sabri, et al. "The role of pineal gland volume in the development of scoliosis." European Spine Journal 32.1 (2023): 181-189.). The 12-17 age range in our study is relatively more restricted.

Additionally, our MUD group had an average age of 16.6 years, the control group 14.8 years, and the athlete group 13.5 years. Based on our previous work and existing literature, we might have expected the MUD group to have a larger brain volume. However, our results indicate the opposite.

We hope that our new analyses, the revised statements in the limitations section, and our explanations help to address your concerns. Thank you again for your valuable critique.

Reviewer 1: Questionable Appropriateness of the Exercise Group as a Control:

The exercise group consisted of enrolled athletes, whose lifestyles (regular high-intensity exercise, dietary habits, and sleep patterns) markedly differ from typical healthy adolescents. Using such a group as the primary control for the MA-dependent group severely hampers the effective separation of "exercise effects" from "MA effects," thereby reducing the internal validity of the study.

Author’s response: Dear Reviewer 1,

Thank you for your valuable feedback.

The exercise group was included to evaluate the potential neuroprotective or modulating effects of regular physical activity on the brain in comparison to the effects of substance use disorder. However, we acknowledge that lifestyle factors (e.g., diet, sleep) in this group might differ from the general healthy population. This difference poses a limitation, making it challenging to isolate the specific effects of exercise from other lifestyle factors.

It's important to clarify that our study does not present the exercise group as a control group for the MUD group. Instead, we compared the effects of MA and exercise on adolescent brain volume. We also included a control group that was not associated with either MA use or exercise to facilitate these comparisons. Our aim was to explain the effects of both sports and MA by comparing statements such as "While the brain structure volume of adolescents with MUD is..., that of exercising adolescents is..."

To provide readers with a clearer understanding of both the athlete and MUD groups, we've included Table 1. This table details the quantity and frequency of substance use among adolescents with MUD, as well as the amount and frequency of sports activity among adolescent athletes.

We have explicitly detailed the limitations concerning factors like IQ and other lifestyle variables (e.g., calorie intake, diet, sleep duration, BMI, step count, age, sex) in the limitations section of our manuscript. We regret that we were unable to achieve optimal control conditions across these difficult-to-study groups, such as adolescents with MUD and licensed adolescent athletes.

Thank you again for your insightful critiques.

***

Reviewer 1: Insufficient Sample Size and Low Statistical Power:

With each group containing only around 10 participants, the sample size is critically inadequate. Such limited numbers significantly reduce the statistical power, making robust conclusions difficult to achieve even with corrections for multiple comparisons (e.g., Bonferroni correction). Thus, the reliability of the study findings remains uncertain.

Inadequate Control of Potential Confounding Factors:

The study lacks adequate control or detailed descriptions of critical variables potentially influencing brain structure, such as intelligence quotient (IQ), lifestyle factors (eating habits, sleep quality, stress levels), and detailed states prior to MRI scanning (e.g., whether the exercise group was scanned immediately post-exercise, drug metabolic conditions). These uncontrolled factors may represent significant confounders, limiting both the reliability and generalizability of the findings.

Overly Optimistic Interpretation of Results Lacking Sufficient Evidence:

The authors speculate that MA-induced reductions in gray matter and increases in white matter may reflect a "compensatory mechanism." However, this interpretation is overly speculative due to a lack of supporting neuropathological or additional neuroimaging evidence (e.g., functional imaging or diffusion imaging).

Author’s response: Dear Reviewer 1,

As you pointed out, making comments about white matter and gray matter without DTI and fMRI studies can only be an assertion. We apologize for the resulting complexity. Following your suggestion, we have presented this statement in our study more clearly, as shown below. We hope this rectifies the situation.

In discussion: “This observation may support the hypothesis that the increase in white matter could be a compensatory response to offset the decrease in gray matter; however, additional evidence corroborated by functional MRI (fMRI) and diffusion tensor imaging (DTI) studies is required to confirm this phenomenon.”

In conclusion: “Our study posits the hypothesis that the observed decrease in gray matter volume in individuals with MUD might be compensated by an increase in white matter volume; however, this hypothesis warrants more detailed investigation with fMRI and DTI studies.”

We also want to emphasize the inherent difficulties in including adolescents with substance use disorder in research (due to its illegal nature and reluctance to participate), indicating that the limitation in sample size was unavoidable. To address the criticism of "low statistical power", we have presented the "observed power" values for our significant findings in our tables.

Thank you for your critiques.

Reviewer 2 Report

Comments and Suggestions for Authors

The article entitled: " The Effects of Active Methamphetamine Use Disorder and  Regular Sports Activities on Brain Volumes in Adolescents". The study's findings are valuable; however, in each section several issues need to be significantly addressed.

Abstract:

The abstract (background) should highlight the role of regular exercise in promoting or preserving brain volume, as this is a key factor distinguishing the athlete group with MUD.

The aim of the study should be explicitly added to establish a clear research focus and enhance the clarity of the study’s objectives and outcomes

Method: mention the setting and gender of participants.

The results should be presented by comparing the MUD group and the athlete group with the control group, followed by a direct comparison between the MUD and athlete groups to highlight specific differences.

Introduction:

  • Page 2 Line 39: (recognized as the second most prevalent illicit drug worldwide) specify the year of this prevalence.
  • Page 2 Line 47: (MA use disorder (MUD) please modify into methamphetamine use disorder (MUD)
  • The potential physiological or molecular mechanisms through which physical activity exerts its neuroprotective effects should be discussed.

Materials and Methods:

  • In Participant Selection, there is a gender imbalance within the groups, particularly in the MUD group (9 males, 1 female) and the runners group (3 males, 7 females). This imbalance could introduce gender-based bias, especially in neurodevelopmental or behavioral outcomes. Consider addressing this limitation or explaining why it was unavoidable.
  • Why were adolescents with active MUD who engage in regular exercise not included in the current study? Including such a group could offer significant insight into the potential protective effects of regular exercise on substance-related outcomes.
  • It is unclear how the authors confirmed that the athlete group was not using any drugs, as the manuscript only states that they refrained from cigarettes and alcohol and had no history of psychosis."

Results

  • Table 1 adds the MA abbreviation in the footnote of the table.
  • It is recommended to restructure the Results section by integrating Figures 1 through 4 within the text, ensuring each figure is properly cited and accompanied by relevant interpretive comments
  • Add study recommendations in the conclusion
  • Abbreviations list: The abbreviations list is missing; please add it.

Author Response

Dear Reviewer 2,

Thank you for your constructive feedback and valuable suggestions, which have significantly contributed to the improvement of our manuscript. We have addressed all your comments point-by-point, and the corresponding changes have been highlighted in yellow within the revised manuscript.

As line numbers are not displayed in the Word version of the journal's template, we were unable to refer to specific line numbers for our explanations. Instead, all revisions and our responses are clearly detailed under each "Author's Response" heading.

We appreciate your thorough review.

Reviewer 2: The article entitled: " The Effects of Active Methamphetamine Use Disorder and  Regular Sports Activities on Brain Volumes in Adolescents". The study's findings are valuable; however, in each section several issues need to be significantly addressed.

Abstract:

The abstract (background) should highlight the role of regular exercise in promoting or preserving brain volume, as this is a key factor distinguishing the athlete group with MUD.

The aim of the study should be explicitly added to establish a clear research focus and enhance the clarity of the study’s objectives and outcomes

Method: mention the setting and gender of participants.

The results should be presented by comparing the MUD group and the athlete group with the control group, followed by a direct comparison between the MUD and athlete groups to highlight specific differences.

Author’s response: Dear Reviewer 2,

Following your suggestions, we have added two sentences to the Abstract regarding the effects of MUD and exercise on the brain. We have also presented the study's aim in a single sentence. We have reorganized the Methods section for greater clarity, including details such as participant gender and the MRI scanner's Tesla strength. Furthermore, we have presented the Results section in more detail. The revised abstract is provided below.

“Abstract: Objective: Methamphetamine abuse during adolescence can significantly im-pact brain development. Conversely, regular exercise is known to support brain health and may possess neuroprotective effects. The aim of this study is to comparatively in-vestigate brain volumes across three distinct adolescent groups: those with active methamphetamine use disorder (MUD), adolescent athletes engaged in regular exercise, and healthy control adolescents. Methods: This MRI study was conducted on three groups of adolescents: 10 with active MUD (9 males, 1 female), 9 licensed runner ado-lescents (3 males, 6 females) , and 10 healthy adolescents (5 males, 5 females). Brain volumes were examined using T1-weighted images obtained from a 3.0 Tesla MRI scanner, followed by automated segmentation via vol2Brain. For statistical analyses, AN-COVA with sex as a covariate and LSD post-hoc tests were performed using SPSS Statis-tics 23. Results: The findings of this study indicate that adolescents with MUDdemon-strated a 10% increase in total white matter volume relative to the athlete group. Con-versely, cortical gray matter volume was notably reduced in the MUD group, specifically by 4% compared to the healthy control group and by 7% compared to the athlete group. Furthermore, volumes of the frontal and insular cortices in the MUD group were signif-icantly diminished when contrasted with the athlete group. In general, individuals with MUD presented with decreased gray matter volumes and increased white matter volumes in their brains. Importantly, the brain volumetric disparities between the MUD group and the athlete group were statistically more significant. Conclusions: In the brains of individuals with MUD, the volume of gray matter areas decreased, while white matter areas increased. These findings reveal that methamphetamine harms the developing adolescent brain. Volumetric differences between the MUD group and the athlete group were observed to be statistically more significant. Therefore, exercise may be associated with potential neuroprotective factor. Further research is needed to explore the potential of exercise-based interventions in mitigating the detrimental effects of methamphetamine abuse.”

Reviewer 2: Introduction:

Page 2 Line 39: (recognized as the second most prevalent illicit drug worldwide) specify the year of this prevalence.

Page 2 Line 47: (MA use disorder (MUD) please modify into methamphetamine use disorder (MUD)

The potential physiological or molecular mechanisms through which physical activity exerts its neuroprotective effects should be discussed.

Author’s response: Dear Reviewer 2,

Thank you for your suggestions. We've incorporated both the year and the initial illicit drug into the text, and we've standardized "MA" to methamphetamine use disorder (MUD) throughout.

As per your recommendation, we've also added information about the neuroprotective mechanisms of exercise to the Introduction section, as detailed below. And new references added in reference section;

“Exercise promotes neuronal survival and neuroplasticity by increasing the production of neurotrophic factors, neurotransmitters, and hormones. Simultaneously, it exerts protective effects on the nervous system by stimulating processes such as synaptic plasticity, neurogenesis, and angiogenesis, while also reducing neuroinflammation. These neuroprotective effects are mediated through enhanced cerebral blood flow, diminished inflammation, and overall improved neuroplasticity. These processes occur through the increased production of key molecules like brain-derived neurotrophic factor and the promotion of exercise-induced myokine release, such as irisin. Additionally, exercise maintains cellular homeostasis by enhancing waste clearance systems like the glymphatic system and autophagy, augmenting antioxidant capacity, and strengthening DNA repair mechanisms”

“10.     Mahalakshmi, B.; Maurya, N.; Lee, S.D.; Bharath Kumar, V. Possible Neuroprotective Mechanisms of Physical Exercise in Neurodegeneration. International journal of molecular sciences 2020, 21, doi:10.3390/ijms21165895.

  1. Tari, A.R.; Walker, T.L.; Huuha, A.M.; Sando, S.B.; Wisloff, U. Neuroprotective mechanisms of exercise and the importance of fitness for healthy brain ageing. Lancet 2025, 405, 1093-1118, doi:10.1016/S0140-6736(25)00184-9.”

Reviewer 2: Materials and Methods:

In Participant Selection, there is a gender imbalance within the groups, particularly in the MUD group (9 males, 1 female) and the runners group (3 males, 7 females). This imbalance could introduce gender-based bias, especially in neurodevelopmental or behavioral outcomes. Consider addressing this limitation or explaining why it was unavoidable.

Why were adolescents with active MUD who engage in regular exercise not included in the current study? Including such a group could offer significant insight into the potential protective effects of regular exercise on substance-related outcomes.

It is unclear how the authors confirmed that the athlete group was not using any drugs, as the manuscript only states that they refrained from cigarettes and alcohol and had no history of psychosis."

Author’s response: Dear Reviewer 2,

We understand your concerns regarding participant selection. The imbalance in the MUD and Athlete groups is an important limitation of our study, and this has been explicitly addressed in the limitations section. To mitigate potential gender-based bias, we have re-analyzed our data. We have now included ANCOVA tests with gender as a covariate, along with observed power and partial eta-squared scores. Notably, a significant portion of the analysis parameters in Table 3 demonstrate high reliability. Furthermore, to account for the influence of gender and other factors in our analyses, we utilized relative volume instead of absolute volume. We hope these new analyses alleviate your concerns.

Recruiting individuals with MUD was particularly challenging for us. Methamphetamine, like in many other parts of the world, is illegal in our country. In Turkey, MA users can receive treatment at the Alcohol and Drug Addiction Treatment and Research Center (AMATEM) and Bakırköy Mental Health and Neurological Diseases Training and Research Hospital. If we had recruited MUD individuals from AMATEM, we would have been studying individuals undergoing MA withdrawal, similar to other studies in the literature. Including adolescents with active MUD in a scientific study requires intense effort. We plan to address individuals with both MUD and athletic involvement in future studies.

Another point to emphasize is that all athletes participating in our study were licensed athletes from Kayseri, within the Turkish Republic Ministry of Youth and Sports. All these athletes regularly participate in competitions. As in other countries, athletes are required to undergo medical examinations and provide health reports before each competition. If substance or drug use is detected in these reports, competitors are disqualified from the competition. We also want to state that all participants in the study (both the control and MUD groups, as well as the athlete group) were screened for drug use, medication use, and all related processes by an experienced psychiatrist.

Reviewer 2: Results

Table 1 adds the MA abbreviation in the footnote of the table.

It is recommended to restructure the Results section by integrating Figures 1 through 4 within the text, ensuring each figure is properly cited and accompanied by relevant interpretive comments

Add study recommendations in the conclusion

Abbreviations list: The abbreviations list is missing; please add it.

Author’s response: Dear Reviewer 2,

We have updated the tables. Therefore, we have provided the explanation for MUD in the new Table 2 and Table 3. In the new analyses, since it was not possible to add partial eta-squared scores and observed power to the GraphPad figures, we have removed the figures created with GraphPad from the study. As per your suggestion, we have added the text below to the conclusions section of the study, providing recommendations for future research.

“Future studies should investigate the potential of exercise-based interventions in mitigating the detrimental effects of MUD. By not limiting research solely to brain volumes, but instead employing advanced neuroimaging and biomarker analysis methods—including functional connectivity, metabolic changes, and neuroinflammation markers—a more comprehensive understanding of the effects of both exercise and MA use on the brain may be achieved.”

The abbreviations list is added.

Thank you for your valuable suggestions and for evaluating our work. Your contributions have significantly helped us and improved our manuscript.

Reviewer 3 Report

Comments and Suggestions for Authors

An excellently conceived and executed study, with well-chosen groups, but I have some concerns.

In Introduction, “Recognized as the second most prevalent illicit drug worldwide [2],” second after which other drug?

In Materials and Methods, “with no history of substance dependence”, better if “with no history of substance use disorder”.

Your groups were different for sex distribution. You recognised this among your Limitations, but did you use sex as a covariate? If not, do it, if yes, write the results.

Was there an ethical committee that approved the study? Could you provide the code number of the approval?

The Conclusions are too lengthy, you could pass some of their material to the end of Discussion and leave just one or two paragraphs.

cm3 should have 3 at its apex.

References 20 and 23-31 lack journal names; all lack co-authors’ names and initials and are not put according to journal guidelines. Journal names, when appearing, are either abbreviated or extended, please be consistent and apply journal instructions to authors.

Author Response

Dear Reviewer 3,

Thank you for your constructive feedback and valuable suggestions, which have significantly contributed to the improvement of our manuscript. We have addressed all your comments point-by-point, and the corresponding changes have been highlighted in yellow within the revised manuscript.

As line numbers are not displayed in the Word version of the journal's template, we were unable to refer to specific line numbers for our explanations. Instead, all revisions and our responses are clearly detailed under each "Author's Response" heading.

We appreciate your thorough review.

Reviewer 3: An excellently conceived and executed study, with well-chosen groups, but I have some concerns.

In Introduction, “Recognized as the second most prevalent illicit drug worldwide [2],” second after which other drug?

Author’s response: Dear Reviewer 3,

Thank you for your question. Your inquiry has led to a significant improvement in our manuscript. In the initial version of the manuscript, we attempted to provide information about MA use by referencing a morphometry study by Farnia et al., which was not the correct approach.

We have now referenced the 2024 report by the United Nations Office on Drugs and Crime, where we found that the majority of countries report cannabis as the most used substance, followed by heroin and then MA. Therefore, we have revised the sentence and the reference to state that cannabis precedes MA.

Thank you for your feedback.

“Recognized as the second most prevalent illicit drug worldwide after cannabis in 2023 [2], “

“2.       Drugs, U.N.O.o. World Drug Report 2024 (Set of 3 Booklets); Stylus Publishing, LLC: 2024.”

Reviewer 3: In Materials and Methods, “with no history of substance dependence”, better if “with no history of substance use disorder”.

Author’s response: Dear Reviewer 3,

We have revised the relevant text as suggested. Thank you for your feedback.

Reviewer 3: Your groups were different for sex distribution. You recognised this among your Limitations, but did you use sex as a covariate? If not, do it, if yes, write the results.

Author’s response: Dear Reviewer 3,

We've re-run our analyses, incorporating sex as a covariate as you suggested. To address concerns about potential bias from the small sample size, we've also included observed power and partial eta-squared scores in our results. Thank you for your valuable feedback.

Reviewer 3: Was there an ethical committee that approved the study? Could you provide the code number of the approval?

Author’s response: Dear Reviewer,

We had already included the detailed information regarding the ethics committee and approval number in the "Institutional Review Board Statement" section. However, following your suggestion, we've also added this information to the Materials and Methods section (see below).

“This study was conducted with the approval of the Cappadocia University Local Ethics Committee, as per decision number 23.18, dated December 27, 2023. Written and oral informed consent was obtained from all participants and their parents or legal guardians prior to the study commencement, adhering to the principles outlined in the Declaration of Helsinki and other pertinent ethical guidelines.”

Reviewer 3: The Conclusions are too lengthy, you could pass some of their material to the end of Discussion and leave just one or two paragraphs.

Author’s response: Dear Reviewer 3,

We understand your suggestion to further reduce the Conclusions section. However, as a result of feedback from other reviewers, we've already added approximately 300 words to the Discussions section.

Given the importance of our study's findings and the need to propose directions for future research, we believe that any further reduction to the Conclusions would diminish the essence and spirit of our work. We feel that a more concise version would inadequately reflect the scope and implications of our study. We regret that we couldn't meet this specific request.

Reviewer 3: cm3 should have 3 at its apex.

References 20 and 23-31 lack journal names; all lack co-authors’ names and initials and are not put according to journal guidelines. Journal names, when appearing, are either abbreviated or extended, please be consistent and apply journal instructions to authors.

Author’s response: Dear Reviewer 3,

We've updated the units and references (added all author, doi, etc information) as per your suggestions. Thank you for your valuable contributions and positive support.

Reviewer 4 Report

Comments and Suggestions for Authors

Your manuscript addresses an important and understudied question—the neuro-anatomical impact of active methamphetamine use during adolescence and the potential neuroprotective role of regular exercise. The combination of volumetric MRI and a three-arm comparative design is commendable. Nevertheless, several issues need attention before the work can be considered for publication.

First, the sample size is very small (10 MUD, 10 athletes, 9 controls) and the sex distribution is markedly unbalanced (MUD ≈ 90 % male; athletes ≈ 70 % female). These factors seriously limit statistical power and external validity. Please provide a post-hoc power calculation and discuss more explicitly how sex and other biological covariates (e.g., age, BMI, pubertal stage) were handled in the analyses. If possible, add sex-adjusted models or at least report effect sizes stratified by sex. 

Second, you report that the data were normally distributed but still chose non-parametric tests because of the limited sample size. While this is defensible, readers would benefit from seeing parametric results (with confidence intervals) alongside Mann–Whitney/Kruskal–Wallis outcomes, or permutation-based tests that retain exact α control while maximizing power. In any case, please add standardized effect sizes (e.g., Cohen’s d, η²) and their 95 % CIs, and consider controlling the experiment-wise error rate with FDR rather than successive Bonferroni corrections, given the very large number of regions assessed 

Third, it is unclear how the athlete group’s training load and cardiorespiratory fitness were quantified. “Licensed runner” status alone does not guarantee comparable exercise exposure across participants. Please specify weekly mileage, training frequency, and competitive level, or provide an objective fitness measure (VOâ‚‚max, beep-test stage, etc.). Without such information, attributing volumetric differences to “exercise” rather than unmeasured lifestyle factors is speculative.

Fourth, the definition of “active” methamphetamine use should be tightened. The inclusion criterion of ≥ 12 months of regular use is helpful, but recency of last intake, cumulative dose, concomitant nicotine/alcohol exposure, and urine toxicology results should be reported. Such details are crucial for interpreting whether observed white-matter enlargement reflects neurotoxicity, neuro-inflammation, or residual vasogenic edema. 

Fifth, please clarify why only relative volumes (%) were emphasized, whereas Table 1 lists absolute volumes that show no significant group effect. Since intracranial volume did not differ significantly between groups, absolute values remain informative and can be used for comparison with prior studies. Consider presenting both metrics consistently in the same table and relocating the long percentage table to supplementary material.

Sixth, several figures could be streamlined to improve readability. Axes labels should include units (e.g., “relative volume [% of ICV]”). In Figures 1–3, annotate the group medians or means directly on the boxplots, and use uniform lettering (a, b, c) instead of mixed superscripts to denote post-hoc differences. Enlarging font sizes and reducing the number of decimal places will also help readers. 

Seventh, in the Discussion, temper causal language. The data are cross-sectional, so statements such as “exercise may protect the adolescent brain” should be rephrased to “exercise might be associated with neuroprotective differences.” Please also add a brief comparison with longitudinal studies where gray-matter recovery after abstinence has been demonstrated, and explain how your findings extend (or contradict) those results. 

Eighth, the English requires moderate refinement for flow and precision. Examples include missing articles (“increase in white matter volume compared to athletes” → “an increase”) and inconsistent tense in the Methods. A thorough language edit by a native speaker or professional service is recommended.

Finally, check minor points:

Spell out all abbreviations at first use (e.g., ICV, WM, GM).

Ensure consistency in units: mm³ (not mm³) throughout tables and text.

Move the funding statement ahead of acknowledgments to match JCM style.

Verify reference formatting; some DOI links are missing.

Addressing these points will strengthen the methodological transparency and interpretability of your study and, in our view, can be achieved with minor to moderate revision.

Comments on the Quality of English Language

The manuscript is generally readable; however, the quality of English can be improved to enhance clarity and fluency. There are issues with article use, verb tense consistency, and sentence structure in several sections, particularly in the Methods and Discussion sections. A professional language edit or review by a native English speaker is recommended to ensure that the scientific content is communicated precisely and fluently.

Author Response

Dear Reviewer 4,

Thank you for your constructive feedback and valuable suggestions, which have significantly contributed to the improvement of our manuscript. We have addressed all your comments point-by-point, and the corresponding changes have been highlighted in yellow within the revised manuscript.

As line numbers are not displayed in the Word version of the journal's template, we were unable to refer to specific line numbers for our explanations. Instead, all revisions and our responses are clearly detailed under each "Author's Response" heading.

We appreciate your thorough review.

Reviewer 4: Your manuscript addresses an important and understudied question—the neuro-anatomical impact of active methamphetamine use during adolescence and the potential neuroprotective role of regular exercise. The combination of volumetric MRI and a three-arm comparative design is commendable. Nevertheless, several issues need attention before the work can be considered for publication.

Reviewer 4: First, the sample size is very small (10 MUD, 10 athletes, 9 controls) and the sex distribution is markedly unbalanced (MUD ≈ 90 % male; athletes ≈ 70 % female). These factors seriously limit statistical power and external validity. Please provide a post-hoc power calculation and discuss more explicitly how sex and other biological covariates (e.g., age, BMI, pubertal stage) were handled in the analyses. If possible, add sex-adjusted models or at least report effect sizes stratified by sex. 

Author’s response: Thank you for your valuable suggestions. We acknowledge that the small sample size and unbalanced gender distribution are significant limitations affecting the statistical power and generalizability of our findings. Therefore, we have thoroughly revised the "Limitations" section of the manuscript (see below).

In response to your feedback, we've re-analyzed the statistical data. We now include partial eta squared values and observed power in the tables to clearly indicate the power of our results. We were particularly pleased to observe that, despite the small sample size, several brain regions (e.g., white matter, gray matter, cortical gray matter) exhibited very high observed power (e.g., 0.960, 0.991).

To minimize the potential influence of gender and other confounding factors, we initially opted to use relative volume (volume of the examined brain region / total intracranial volume [white matter + gray matter + CSF]) instead of absolute volume. We've clarified this methodological choice and re-explained it in the revised "Limitations" section (see below).

In accordance with your recommendation, we have re-performed the ANCOVA analysis, incorporating gender as a covariate.

We recognize that age is an important covariate. However, in the literature on adolescent brain studies, there are examples that consider the 10-18 age range as a single group (Batın, Sabri, et al. "The role of pineal gland volume in the development of scoliosis."European Spine Journal 32.1 (2023): 181-189.). The 12-17 age range is relatively more restricted. Additionally, our MUD group had an average age of 16.6 years, the control group 14.8 years, and the athlete group 13.5 years. Based on our previous work and existing literature, we might have expected the MUD group to have a larger brain volume. However, our results indicate the opposite.

Your insightful suggestions have significantly contributed to the improvement of our study, and we are truly grateful for your valuable input.

 “A significant limitation of the current study stems from the inability to comprehen-sively characterize the participants' diet, lifestyle, sex, intelligence quotient, and specific phenotypes. These unmeasured variables may have exerted an influence on the observed volumetric differences in brain structures, potentially confounding the interpretation of our findings. Future research should endeavor to incorporate comprehensive assessments of these factors to elucidate their potential moderating or mediating effects on brain morphology. Furthermore, the sample size in this study was notably small, comprising a limited number of participants with MUD and athlete adolescents. This inherent small sample size significantly restricts the generalizability of our findings. While the illicit nature of methamphetamine, the inherent challenges in identifying adolescents with MUD, and their understandable reluctance to participate in research pose considerable obstacles to recruitment, larger-scale investigations are critically needed to enhance the robustness and external validity of future research in this area. An additional constraint of our study was the heterogeneous gender distribution within the participant groups. A substantial majority (90%) of the adolescents with MUD were male, while majorities (66.6%) of the athlete adolescents were female. To mitigate the potential confounding effect of gender and age on brain volume, we opted to analyze relative brain volumes (%=mm3/ICV) rather than absolute volumes (mm3). Nevertheless, future studies should strive for more balanced gender representation to enable more nuanced investigations into potential sex-specific differences in brain structure among these populations. Longitudinal studies employing multi-modal assessments that concurrently analyze di-etary habits, age-related changes, and specific phenotypic characteristics are warranted to more comprehensively unravel the intricate effects of MUD on brain structure.”

Reviewer 4: Second, you report that the data were normally distributed but still chose non-parametric tests because of the limited sample size. While this is defensible, readers would benefit from seeing parametric results (with confidence intervals) alongside Mann–Whitney/Kruskal–Wallis outcomes, or permutation-based tests that retain exact α control while maximizing power. In any case, please add standardized effect sizes (e.g., Cohen’s d, η²) and their 95 % CIs, and consider controlling the experiment-wise error rate with FDR rather than successive Bonferroni corrections, given the very large number of regions assessed 

Author’s response: Thank you for your valuable feedback. We have revised our statistical analyses as you suggested and rewritten the "Statistical Analysis" section accordingly (see below). In both new Table 2 and Table 3, we've now reported the partial eta-squared (ηp2​) values, indicating the proportion of variance explained. These ηp2​ values are provided as a measure of effect size. We acknowledge that using LSD for exploratory purposes, especially with 135 regions, increases the risk of Type I error. Future studies with larger sample sizes could certainly employ more stringent multiple comparison corrections, such as FDR.

Reviewer 4: Third, it is unclear how the athlete group’s training load and cardiorespiratory fitness were quantified. “Licensed runner” status alone does not guarantee comparable exercise exposure across participants. Please specify weekly mileage, training frequency, and competitive level, or provide an objective fitness measure (VOâ‚‚max, beep-test stage, etc.). Without such information, attributing volumetric differences to “exercise” rather than unmeasured lifestyle factors is speculative.

Fourth, the definition of “active” methamphetamine use should be tightened. The inclusion criterion of ≥ 12 months of regular use is helpful, but recency of last intake, cumulative dose, concomitant nicotine/alcohol exposure, and urine toxicology results should be reported. Such details are crucial for interpreting whether observed white-matter enlargement reflects neurotoxicity, neuro-inflammation, or residual vasogenic edema. 

Author’s response: Dear Reviewer 4,

Thank you for your valuable feedback. We'd like to clarify that all participants included in our study are licensed athletes and engage in the same sport (running). While we did not collect objective data such as VO2 or fitness measurements from the athletes, nor did we gather data on urine test results for individuals with MUD, we have taken steps to present more objective information about the groups.

To this end, we've prepared a new table (Table 1. Detailed Characteristics of Licensed Athlete and Methamphetamine Use Disorder Groups) that provides additional details for readers. This table now includes information on athletes' weekly training duration and distance covered, as well as the daily and weekly MA dosages and frequency of use for individuals with MUD. We hope this added information addresses your concerns and provides a more comprehensive overview for the readers.

Thank you again for your constructive contribution.

Reviewer 4: Fifth, please clarify why only relative volumes (%) were emphasized, whereas Table 1 lists absolute volumes that show no significant group effect. Since intracranial volume did not differ significantly between groups, absolute values remain informative and can be used for comparison with prior studies. Consider presenting both metrics consistently in the same table and relocating the long percentage table to supplementary material.

Author’s response: Dear Reviewer 4,

Thank you for your valuable feedback. We'd like to explain why we chose not to present absolute volumetric data without incorporating total intracranial volume (ICV).

One of the main challenges in interpreting brain volume data is the significant individual variability in overall brain and head size across the human population. This natural variation makes direct comparisons of absolute brain region volumes between individuals or groups quite complex. When trying to determine if a brain structure is "smaller" or "larger," these individual differences can lead to complications. This isn't just a methodological choice for us; it's essential for achieving meaningful biological results. If we don't account for these natural individual differences in overall brain size, observed regional volume differences in a patient group or specific condition could be mistakenly attributed to disease-specific pathology, rather than pre-existing size differences among individuals. This could lead to incorrect conclusions and potentially flawed diagnoses. Therefore, addressing individual variability through relative measurements is a fundamental prerequisite for the clinical and scientific validity of volumetric neuroimaging.

Absolute brain volume refers to the raw, unadjusted size of the brain or its substructures. Historically, absolute brain size was sometimes considered a direct measure of cognitive capacity or intelligence. However, this view has largely been abandoned due to its oversimplification and lack of validity. Researchers like Alexander Brandt and Otto Snell, even in the 1800s, showed the importance of scaling brain size to body size, marking the end of using absolute brain size as a valid measure of brain capacity.

For example, macrocephaly (large head size) is commonly reported in children with autism, and MRI studies have found larger brains in autistic children compared to controls. If no adjustment is made for brain size, this might be interpreted as a global increase in brain volumes in autism (meaning all structures are proportionally larger). However, adjustments help us more accurately determine if brain overgrowth is due to regionally specific enlargements of certain brain structures. This highlights a critical methodological pitfall where apparent global growth can mask specific regional abnormalities. Normalization ensures that research findings are biologically meaningful and aren't just statistical artifacts of head size differences across populations (as discussed in https://pmc.ncbi.nlm.nih.gov/articles/PMC3510982/).

Furthermore, we observed that the absolute brain volumes presented in the old Table 1 (now Table 2 in the revised version) did not have high accuracy for either partial eta-squared scores or observed power values. For this reason, along with the points mentioned above, we chose to present relative values in our study.

We hope this addresses your critique. We also want to emphasize that your criticisms have broadened our perspective. Thank you for your contributions.

***

Reviewer 4: Sixth, several figures could be streamlined to improve readability. Axes labels should include units (e.g., “relative volume [% of ICV]”). In Figures 1–3, annotate the group medians or means directly on the boxplots, and use uniform lettering (a, b, c) instead of mixed superscripts to denote post-hoc differences. Enlarging font sizes and reducing the number of decimal places will also help readers. 

Author’s response: Dear Reviewer,

Thank you for your suggestions regarding the figure improvements. We understand your point, but with our new analyses, we've now included both partial eta-squared scores and observed power values in our tables. Additionally, we used LSD tests for group comparisons.

We're unsure if our current software, GraphPad Prism, can simultaneously present all these data within the figures. Therefore, instead of replicating the table information in graphs, we've decided to remove the figures containing graphs from the manuscript.

Thank you again for your valuable suggestions.

Reviewer 4: Seventh, in the Discussion, temper causal language. The data are cross-sectional, so statements such as “exercise may protect the adolescent brain” should be rephrased to “exercise might be associated with neuroprotective differences.” Please also add a brief comparison with longitudinal studies where gray-matter recovery after abstinence has been demonstrated, and explain how your findings extend (or contradict) those results. 

Author’s response: Dear Reviewer 4,

We've addressed your feedback regarding the causal language used in the Discussion and Conclusion sections. We have softened the phrasing to reflect a less definitive tone. All revisions are presented below.

"this finding suggests that methamphetamine use may lead to an increase in white matter volume compared to exercise.": The key change here is the inclusion of "may lead to".

"These results underscore a statistically significant effect of MA exposure during adolescence on the continuously developing brain." was changed to "These results underscore a statistically significant association of MA exposure during adolescence with alterations in the continuously developing brain."

"This may indicate that sport increases cortical thickness naturally rather than through glial proliferation or neuronal damage." was changed to "This may suggest that sport is associated with greater cortical thickness, potentially reflecting natural development rather than processes like glial proliferation or neuronal damage."

"It also suggests that glial proliferation is suppressed during adolescence." was changed to "It also suggests that glial proliferation may be modulated during adolescence."

"This may indicate that adolescents with MUD will be more vulnerable to gray matter deficits as they age." was changed to "This may suggest that adolescents with MUD could be more vulnerable to gray matter deficits as they age."

"Therefore, incorporating regular exercise into the treatment regimen for MA use disorder could positively influence the therapeutic trajectory, potentially leading to more lasting behavioral modifications within a shorter timeframe." was changed to "Therefore, incorporating regular exercise into the treatment regimen for MA use disorder may be associated with a more positive influence on the therapeutic trajectory, potentially leading to more lasting behavioral modifications within a shorter timeframe."

"suggesting that MA use may lead to more pronounced effects on these structures in later life." was changed to "suggesting that MA use may be associated with more pronounced effects on these structures in later life."

"could enhance therapeutic outcomes and expedite behavioral modifications." was changed to "could potentially enhance therapeutic outcomes and expedite behavioral modifications."

Thanks for your suggestion. We've added a paragraph to the Discussion section, as you recommended, comparing our findings—especially regarding gray matter—with other studies. This new section is concise, keeping to under a single paragraph.

We want to note that we couldn't go into extensive detail with these comparisons. The main reason is that most other studies don't focus on individuals with active MUD, and their participant age ranges differ significantly from the adolescents in our study.

The added paragraph and references are provided below. We appreciate your helpful feedback.

           “Ruan et al. conducted a comparative study involving a cohort of individuals with a mean MA use duration of 6.3 years and a 6-month abstinence period, against a control group of non-substance users. Their findings revealed significantly reduced gray matter volume (GMV) in the precentral gyrus, caudate head, fusiform gyrus, and cerebellum among the 6-month abstinent MA users compared to the drug-naïve controls. Further-more, a longitudinal comparison indicated that GMV was higher in the cerebellum but lower in the cingulate gyrus after 12 months of abstinence, as opposed to 6 months [33]. In a separate investigation, Zhang et al. compared 44 individuals who had abstained from methamphetamine for at least 14 months with 40 healthy controls. The MA group exhibited increased GMV in the bilateral cerebellum but reduced volumes in the right calcarine and right cuneus when contrasted with the healthy control group. Additionally, a positive correlation was observed between GMV in the left cerebellum and the duration of abstinence within the MA group [34]. More recently, Hu et al. examined 48 males with MUD and 66 healthy controls. Their study identified smaller volumes in the right cuneus gyrus, left lingual gyrus, bilateral supramarginal gyrus, right inferior parietal gyrus, right dorsal anterior cingulate cortex, and left nucleus accumbens in individuals with MUD [35]. A notable finding in our current study was the reduced gray matter volume in cor-tical and insular structures, as well as in frontal lobe regions, among individuals with MUD compared to athletes. This distinction in the active substance use status of partici-pants in our study, as opposed to the abstinence-focused cohorts typically examined in existing literature, presents a challenge for direct comparisons with other studies.“

  1. Ruan, X.L.; Zhong, N.; Yang, Z.; Fan, X.D.; Zhuang, W.X.; Du, J.; Jiang, H.F.; Zhao, M. Gray matter volume showed dynamic alterations in methamphetamine users at 6 and 12 months abstinence: A longitudinal voxel-based morphometry study. Prog Neuro-Psychoph 2018, 81, 350-355, doi:10.1016/j.pnpbp.2017.09.004.
  2. Zhang, Z.; He, L.; Huang, S.; Fan, L.; Li, Y.; Li, P.; Zhang, J.; Liu, J.; Yang, R. Alteration of Brain Structure With Long-Term Abstinence of Methamphetamine by Voxel-Based Morphometry. Front Psychiatry 2018, 9, 722, doi:10.3389/fpsyt.2018.00722.
  3. Hu, X.Y.; Jiang, P.; Gao, Y.X.; Sun, J.Y.; Zhou, X.B.; Zhang, L.Q.; Qiu, H.; Li, H.L.; Cao, L.X.; Liu, J.; et al. Brain morphometric abnormalities and their associations with affective symptoms in males with methamphetamine use disorder during abstinence. Front Psychiatry 2022, 13, 1003889, doi:ARTN 1003889, 10.3389/fpsyt.2022.1003889.

Reviewer 4: Eighth, the English requires moderate refinement for flow and precision. Examples include missing articles (“increase in white matter volume compared to athletes” → “an increase”) and inconsistent tense in the Methods. A thorough language edit by a native speaker or professional service is recommended.

Finally, check minor points:

Spell out all abbreviations at first use (e.g., ICV, WM, GM).

Ensure consistency in units: mm³ (not mm³) throughout tables and text.

Move the funding statement ahead of acknowledgments to match JCM style.

Verify reference formatting; some DOI links are missing.

Addressing these points will strengthen the methodological transparency and interpretability of your study and, in our view, can be achieved with minor to moderate revision.

Author’s response: Dear Reviewer 4,

In response to the valuable critiques from you, we have made the following revisions to the manuscript:

  • Abbreviations: We have now included a comprehensive list of abbreviations. Each abbreviation is defined at its first appearance within the text.
  • Units: We have corrected all instances of units, specifically "mm3".
  • Funding Sources: The "Funding Sources" section has been relocated to precede the "Acknowledgments."
  • References: The initial submission contained inconsistencies in reference formatting. We have thoroughly revised all references, striving for accurate presentation of DOIs and authors.

We would also like to express our sincere gratitude for the exceptionally thorough and constructive feedback during this review process; it has significantly enhanced the quality of our work.

Round 2

Reviewer 1 Report

Comments and Suggestions for Authors

 This study investigates brain volume alterations in adolescents with methamphetamine use disorder (MUD) using MRI and automated segmentation, comparing them with adolescent athletes and healthy controls. The topic is highly novel and socially relevant. In particular, the successful recruitment of a rare and representative MUD adolescent sample, along with a physically active control group, significantly enhances the study's scientific and practical value.  The paper is technically well-executed in terms of imaging acquisition, data processing, and statistical analysis. The results clearly demonstrate significant gray matter volume reduction and white matter increase in the MUD group, supporting the authors’ hypothesis regarding the neurotoxic effects of methamphetamine and suggesting a possible neuroprotective role of regular physical activity. However, several limitations in the study design merit attention:1.There are group imbalances in gender and age, which may confound brain volume comparisons;2.The sample size is small, restricting generalizability;3.Potential confounding factors (e.g., IQ, diet, sleep, mental health) were not accounted for. I recommend that the authors more explicitly discuss these limitations and adopt a cautious tone in drawing causal conclusions. If feasible, a brief explanation regarding the representativeness or stability of the sample would further strengthen the manuscript. In conclusion,  some methodological shortcomings, the manuscript’s originality, feasibility, and potential scientific impact justify needs minor-to-moderate revision.

Author Response

Dear Reviewer 1,

Thank you for your valuable feedback and important suggestions, which have significantly contributed to our manuscript. We have addressed your comments point-by-point, and the corresponding changes have been highlighted in yellow within the revised manuscript.

As line numbers are not displayed in the Word version of the journal's template, we were unable to refer to specific line numbers for our explanations. Instead, all revisions and our responses are clearly detailed under each "Author's Response" heading.

We appreciate your thorough review.

***

Reviewer 1: This study investigates brain volume alterations in adolescents with methamphetamine use disorder (MUD) using MRI and automated segmentation, comparing them with adolescent athletes and healthy controls. The topic is highly novel and socially relevant. In particular, the successful recruitment of a rare and representative MUD adolescent sample, along with a physically active control group, significantly enhances the study's scientific and practical value.  The paper is technically well-executed in terms of imaging acquisition, data processing, and statistical analysis. The results clearly demonstrate significant gray matter volume reduction and white matter increase in the MUD group, supporting the authors’ hypothesis regarding the neurotoxic effects of methamphetamine and suggesting a possible neuroprotective role of regular physical activity. However, several limitations in the study design merit attention:1.There are group imbalances in gender and age, which may confound brain volume comparisons;2.The sample size is small, restricting generalizability;3.Potential confounding factors (e.g., IQ, diet, sleep, mental health) were not accounted for. I recommend that the authors more explicitly discuss these limitations and adopt a cautious tone in drawing causal conclusions. If feasible, a brief explanation regarding the representativeness or stability of the sample would further strengthen the manuscript. In conclusion,  some methodological shortcomings, the manuscript’s originality, feasibility, and potential scientific impact justify needs minor-to-moderate revision.

Author’s response: Dear Reviewer 1,

Thank you for your valuable feedback. We have revised the "Limitations" section of our manuscript (see below) to address your comments.

Specifically, we have now explicitly stated that we were unable to comprehensively analyze participant-specific factors such as diet, IQ, and lifestyle. We acknowledge in the limitations section that these unmeasured factors may have influenced the findings.

We have also clarified that the small sample size is a significant limitation and that future studies should consider this, aiming for larger samples. We further explained that the recruitment of adolescent participants presented "inherent challenges," contributing to the limited sample size. The generalizability of our study's findings has been explicitly noted as a restriction.

Furthermore, we have stated that this study serves as a preliminary investigation and emphasized the necessity for future longitudinal studies that take into account other factors such as gender, age, and dietary habits.

We believe these revisions enhance the clarity and robustness of our manuscript, reflecting a cautious tone in drawing conclusions while highlighting areas for future research.

“The current study is limited in its ability to fully characterize participant-specific factors such as diet, lifestyle, intelligence quotient (IQ), and specific phenotypes. These unmeasured variables may have influenced the observed differences in brain structures, complicating the interpretation of the findings. Future research should include compre-hensive assessments of these factors to better understand their potential effects on brain morphology. Additionally, the study had a small sample size with a limited number of participants who were both MUD-affected and adolescent athletes, which significantly limits the generalizability of the results. Larger studies are needed to improve the power and external validity of future research in this area. The gender distribution in the study was imbalanced, with males being the majority in the MUD group and females in the athlete group. To address this, relative brain volumes were analyzed, and gender was included as a covariate in the analysis. This study is a pilot study, and future studies should strive for a more balanced gender distribution and a larger sample size to inves-tigate potential gender-specific differences in brain structure. Longitudinal studies that consider dietary habits, age-related changes, and specific characteristics are necessary to gain a better understanding of the effects of MUD on brain structure.”

Thank you for your valuable feedback regarding the language of our manuscript. We have thoroughly reviewed and revised the entire paper to improve its linguistic quality.

We made a concerted effort to adopt a more cautious and humble tone throughout the manuscript. All grammatical errors have been corrected. Previously, the language was somewhat complex, with unnecessarily long sentences that led to issues in clarity. We also recognized weaknesses in our use of English tenses and suffixes. To address these points, we have rephrased the entire manuscript, replacing lengthy and superfluous sentences with clearer and more concise expressions. We identified and corrected instances of awkward phrasing and structural inaccuracies.

We are grateful for the opportunity you have given us to enhance our manuscript.

Reviewer 2 Report

Comments and Suggestions for Authors

All inquiries were addressed by authors 

Author Response

Reviewer 2: All inquiries were addressed by authors

Author’s Response: Dear Reviewer 2,

Thank you for evaluating our study and providing us with the opportunity for revision, as well as for your contributions to the improvement of our manuscript.
